# New Therapies of Liver Diseases: Hepatic Encephalopathy

**DOI:** 10.3390/jcm10184050

**Published:** 2021-09-07

**Authors:** Chiara Mangini, Sara Montagnese

**Affiliations:** Department of Medicine, University of Padova, 35128 Padova, Italy; chiara.manginij@gmail.com

**Keywords:** cirrhosis, portal-systemic shunt, ammonia, vigilance

## Abstract

Hepatic encephalopathy (HE) is a common complication of advanced liver disease which has profound implications in terms of the patients’ ability to fulfil their family and social roles, to drive and to provide for themselves. Recurrent and persistent HE is still a serious management challenge, translating into a significant burden for patients and their families, health services and society at large. The past few years have been characterized by significantly more attention towards HE and its implications; its definition has been refined and a small number of new drugs/alternative management strategies have become available, while others are underway. In this narrative review we summarize them in a pragmatic and hopefully useful fashion.

## 1. Introduction

Hepatic encephalopathy (HE) is a brain dysfunction caused by liver failure and/or portal-systemic shunt that results in a spectrum of neurological and psychiatric abnormalities ranging from subclinical alterations to coma [1].

Classification is based on the underlying condition leading to HE: “Type A” HE is due to acute liver failure, “Type B” HE is caused by portal-systemic shunt without significant liver disease and “Type C” HE by cirrhosis with or without portal-systemic shunt [1]. In terms of its severity, HE is qualified as covert (minor or no signs/symptoms but abnormalities on neuropsychological and/or neurophysiological tests) or overt (Grades II or over according to the West Haven criteria [1]). Finally, in terms of its time-course, overt HE is classified as episodic, recurrent (more than one episode over a period of six months) or persistent (no return to normal/baseline neuropsychiatric performance in between episodes) [1].

This narrative review will deal with the management of Type C HE, i.e., the one associated with cirrhosis with or without portal-systemic shunt [1]. Standard information on HE treatment (no formal literature search) plus 2018–2021 information on novel therapies is included. The 2018–2021 literature search was conducted on Pubmed using the terms hepatic encephalopathy plus: treatment, polyethylene glycol (PEG), branched-chain amino acids (BCAAs), L-ornithine L-aspartate (LOLA), nitrogen scavengers, Acetyl L-carnitine, albumin, probiotics, fecal microbiota transplantation (FMT), flumazenil, minocycline, ibuprofen, phosphodiesterase-5 inhibitors, indomethacin, benzodiazepine inverse agonists.

Prior to discussing available, recommended and more experimental (Table 1) treatment options, it is important to highlight how:-The HE phenotype is nonspecific and differential diagnosis extremely important;-Response to treatment can be utilized to confirm a working diagnosis of HE, especially in its mild forms;-The lack thereof, should prompt fast differential diagnosis investigations, especially in severe forms;-With the exception of direct modulation of vigilance/inflammation (*vide infra*), HE treatment is essentially synonymous of ammonia-lowering treatment. Hyperammonaemia is necessary but not sufficient for a working diagnosis of HE (i.e., there is no HE without hyperammonaemia but the presence of hyperammonaemia does not necessarily translate into a HE phenotype, especially in young patients) [1].

## 2. General Management Principles

An episode of overt HE is generally managed by ensuring adequate airway protection for severe cases, correction of any identified precipitating factors and institution of ammonia-lowering treatment [1,2]. The most commonly utilized drugs for the subsequent commencement of secondary prophylaxis are non-absorbable disaccharides (i.e., lactulose or lactitol, which aim at reducing gut nitrogen load via their laxative and prebiotic effects, enhancing bacterial ammonia uptake and reducing ammonia production in the small intestine) and non-absorbable antibiotics (also affecting production/absorption of gut-derived neurotoxins, and reducing endotoxemia and inflammation), which are generally added after a second overt HE episode or, as a stand-alone, when non-absorbable disaccharides are not well tolerated [3,4]. Primary prophylaxis is not generally recommended, with the exception of the rapid removal of blood from the gastrointestinal tract after an upper gastrointestinal bleed, for example with lactulose [5] or mannitol [6]. By contrast, secondary prophylaxis is important, as once a patient has experienced an episode of overt HE, the likelihood of further episodes is high and a common cause of re-admission into hospital [7]. Secondary prophylaxis with a non-absorbable disaccharide (lactulose or lactitol) should be instituted [8,9]. If this is adequately titrated and the patient/caregivers instructed carefully, both constipation and diarrhoea/excessive flatulence and abdominal distension can be avoided. If overt HE becomes recurrent (i.e., more than one episode within six months [1]), rifaximin should be added to help maintain remission [1,10].

There are more uncertainties on the benefits of treatment of patients with mild forms of HE, especially as not all centres have the experience to diagnose them. Thus, treatment is not routinely recommended [1]. However, its initiation, especially if “ecological” (i.e., adding soluble fermentable fibre, probiotics such as yogurt, vegetables, cereal and milk-derived proteins to the habitual diet, plus getting used to in-between-meals snacks and a bedtime snack [11]), may both confirm the diagnosis and be beneficial. It is, therefore, reasonable, especially if the patients or their caregivers report symptoms compatible with mild HE, to institute it.

The management of patients with recurrent or persistent HE, which is more common in patients with large, spontaneous or surgical shunts [12] can be very difficult. Shunt closure/reduction can be considered in patients whose shunts are accessible. When related to TIPS, it can be treated and even prevented by reducing or occluding the stent [13,14,15]. Recurrent or persistent overt HE, and those forms of HE which are dominated by motor dysfunction (hepatic myelopathy) often require combination treatment, together with changes in the sources of dietary protein. Branched-chain amino acids (BCAAs), probiotics, L-ornithine L-aspartate (LOLA), non-ureic nitrogen scavengers and albumin have all been tested and can be used within this context [16]. Liver transplantation is the ultimate therapeutic option, and possibly the only real one for hepatic myelopathy [17]. It is crucial that all significant shunts are closed during transplantation, to avoid post-transplant Type B HE [1].

## 3. Therapies Other Than Non-Absorbable Disaccharides and Non-Absorbable Antibiotics

These are summarized in Table 1, at the end of this section.

### 3.1. Polyethylene Glycol (PEG)

The osmotic laxative PEG as a stand-alone [18] or in association with lactulose [19] has been associated with faster resolution of an episode of overt HE requiring hospitalization. Naso-gastric tubes have been used to administer PEG in patients with severe HE to avoid aspiration and guarantee adequate doses. Therefore, PEG applicability may be limited to patients in whom naso-gastric tube placement is safe and successful. A recent review and meta-analysis of four studies [20] showed that a single dose of PEG significantly improved clinical features of HE after 24 h and reduced the number of days for HE resolution compared to lactulose; however, no differences were observed in hospitalization length.

### 3.2. L-Ornithine L-Aspartate (LOLA)

LOLA is a substrate for the urea cycle and increases urea production in peri-portal hepatocytes. In addition, it activates glutamine synthetase in peri-venous hepatocytes and the skeletal muscle [21].

In a 2013 meta-analysis, LOLA was significantly more effective on HE than placebo/no-intervention [22]. Two comparative studies of LOLA and lactulose showed similar efficacy [23,24]. A double-blind randomised controlled trial (RCT) showed that LOLA was superior to placebo in the secondary prophylaxis of overt HE in 150 patients with cirrhosis [25]; improvement in psychometric test scores, critical flicker frequency and quality of life were documented, together with significant reductions in arterial ammonia levels. In one study, LOLA was shown to shorten a bout of overt HE requiring hospitalization when added to non-absorbable disaccharides and ceftriaxone [26]. The decrease in ammonia associated with LOLA seems to be temporary, and rebound hyperammonaemia has been observed on cessation [27]. Intravenous LOLA can be used to treat patients unresponsive to conventional therapy but further research is required in determining treatment dosage and duration. A Cochrane review [28] scored the available evidence in favour of LOLA as very low and qualified its benefits as uncertain, while a recent review and meta-analysis [29] suggests that LOLA is comparable to other ammonia-lowering agents in treating HE of varying severity.

### 3.3. Non Ureic Nitrogen Scavengers

**Sodium benzoate** provides an alternative pathway for nitrogen disposal and has been mostly used to treat patients with urea cycle defects. While the results of one, available RCT in patients with HE were encouraging [30], sodium benzoate has also been associated with increased ammonia levels in standard conditions and after a glutamine challenge [31]. A 2019 Cochrane systematic review did not find significant differences between sodium benzoate and non-absorbable disaccharides in terms of mortality, grade of HE or blood ammonia levels in patients with cirrhosis and an episode of overt HE [32]. Sodium benzoate may be particularly appropriate when HE and hyponatremia coexist [33].

**Sodium phenylbutyrate** may also reduce ammonia levels and improve neurological status/discharge survival in intensive care patients with overt HE [34]

**Glycerol phenylbutyrate** is mostly used in urea cycle disorders, as it favours nitrogen elimination by combining phenylacetic acid (a metabolite of phenylbutyric acid) with glutamine to form phenylacetyl-glutamine, which is then excreted in the urine. In a randomized, double-blind Phase IIb study of patients with cirrhosis who had had an episode of HE and were already on treatment with lactulose and rifaximin [35], glycerol phenylbutyrate significantly reduced ammonia levels and the likelihood of overt HE recurrence.

The rationale for treatment with **ornithine phenylacetate** relates to its capacity to stimulate glutamine synthetase in peripheral organs, by incorporating ammonia into the ‘nontoxic’ molecule phenylacetylglutamine, which is then excreted in the urine [36]. A randomized trial of 38 patients with cirrhosis enrolled within 24 h of an upper gastrointestinal bleed [37] showed that ornithine phenylacetate was well tolerated but did not significantly decrease ammonia. A subsequent RCT [38] of 231 patients did not document significant differences in time to clinical improvement compared to placebo.

### 3.4. Nutrition

Malnutrition, and sarcopenia in particular, are common in patients with cirrhosis and are associated with decreased survival [39]. Muscle loss impinges on nitrogen and ammonia metabolism and is associated with an increased risk of HE [40]. Thus, a low protein diet should be avoided in patients with HE [33,41,42], whose recommended daily energy and protein intake is not different from that of patients with cirrhosis and no HE (i.e., 35–40 kcal/kg ideal body weight and 1.2–1.5 g/kg ideal body weight, respectively). Nutritional therapy has been shown to improve psychometric performance and quality of life and to reduce the risk of overt HE compared with no intervention in patients with minimal HE in one RCT [43].

Small meals, evenly distributed throughout the day, and a late evening snack of complex carbohydrate [44] should be encouraged, as they decrease protein catabolism, and interrupt the long fast between dinner and breakfast. Personalized physical exercise should also be encouraged [45]. Despite solid rationale, there is limited evidence for the advantages of the replacing meat with vegetable/dairy protein. This is recommended only in the minority of patients who are truly intolerant of meat protein, and should be performed by experts and monitored closely to avoid reduction in caloric and protein intake [41].

A meta-analysis of four RCTs [46] showed that zinc supplementation improved some psychometric tests but did not reduce overt HE recurrence.

Diagnosed of suspected deficits in vitamins and micronutrients should be treated, as they may worsen mental function and confound HE diagnosis [33,41].

### 3.5. Albumin

Albumin may be effective in preventing overt HE in patients with decompensated cirrhosis, as demonstrated by a large retrospective study [47] and a recent meta-analysis [48]. In one RCT, high dose albumin was associated with decreased risk of high grade overt HE compared to standard of care in patients with cirrhosis and ascites [49].

### 3.6. Branched-Chain Amino Acids (BCAAs)

BCAAs availability is decreased in patients with cirrhosis, impinging on ammonia to glutamine conversion in the skeletal muscle. Thus, BCAA supplementation may enhance ammonia detoxification and reduce its concentrations. A Cochrane review [50] of 16 RCTs comparing BCAA (oral or intravenous) to placebo, no intervention, diet, neomycin or lactulose documented no beneficial effects of BCAAs on HE when trials including lactulose or neomycin were excluded, and no difference when BCAAs and lactulose or neomycin were compared. No effects on mortality, quality of life, or nutritional parameters was observed either. Recently, a multicenter prospective study evaluated long term oral BCAAs supplementation in decompensated cirrhosis [51]. In this study, MELD and Child-Pugh scores significantly improved in patients supplemented with BCAAs for more than six months; moreover, episodes of overt HE occurred less in the BCAA group compared to the control group. An increase in serum albumin in patients with a low albumin levels during BCAA supplementation was also observed. Furthermore, isoleucine long-term supplementation has been associated with increased brain perfusion and clinical improvement of overt HE compared to leucine [52]. BCAAs can be considered as a way to guarantee adequate protein intake in patients with recurrent/persistent HE on vegetable/diary diets [16]. However, palatability represents a significant issue [53].

### 3.7. Acetyl L-carnitine (ALC)

ALC contributes to blood and brain ammonia reduction and facilitates cellular energy production trough the uptake of acetyl-coenzyme A into the mitochondria, thereby preventing ammonia-induced neurotoxic damage in patients with HE [54].

A meta-analysis of seven RCTs (all single centre and of small to moderate size) [55] including 660 patients with varying degree of HE concluded that ALC reduced ammonia levels and improved one paper and pencil neuropsychological test. A 2019 Cochrane review of five single-centre small RCTs for a total of 398 patients [56] assessed the benefits/harms of ALC in patients with HE compared to intervention, placebo, or standard therapy. No summary information could be obtained on mortality, serious adverse events, or days of hospitalisation, nor were there any obvious differences in terms of fatigue, quality of life and minor adverse events. This may relate to limitations in the design and execution of the trials included. Blood ammonia lowering was documented in patients receiving acetyl-L-carnitine, although it was not associated with any obvious clinical benefit.

### 3.8. Probiotics

The rationale for the use of probiotics in treating HE relates to their capacity to modulate the gut microbiota composition and metabolic function, and to reduce inflammation. A recent review [57] compared the effects of probiotics or symbiotics (a combination of pre- and probiotics) with placebo/no intervention, or with any other treatment in patients with an episode or with persistent HE. A variety of probiotics and symbiotics were used and treatment duration ranged from three weeks to twelve months. Compared with placebo or no intervention, probiotics and symbiotics prevented overt HE recurrence [57]. However, this was not confirmed when probiotics were compared with lactulose, rifaximin or LOLA [57]. Based on a Cochrane review [58], the majority of trials were found to suffer from a high risk of both systematic and random error. A more recent meta-analysis [59] including 14 RCTs and 1132 patients found that probiotics decreased ammonia and endotoxin levels, improved minimal HE, and prevented overt HE. Of interest, the effects of fermentable fibre alone were comparable to those of a symbiotic on mental performance, ammonia levels and the gut flora in patients with minimal HE in one study [60].

### 3.9. Fecal Microbiota Transplantation (FMT)

The rationale for FMT for the treatment of HE is to modulate the composition and function of the gut microbiota. In one RCT, a small group of patients with recurrent HE where either treated with antibiotics and FMT or with standard of care: an improvement in short-term cognitive function and hospitalization was observed. Recently, this study was extended to assess the long-term impact of FMT on cognition, hospitalizations, and HE recurrence [61]: long-term safety and improvement in clinical and cognitive function parameters were confirmed in patients who received FMT with pre-treatment antibiotics compared with standard of care, especially regarding prevention of HE recurrence and hospitalizations. However, larger trials are needed. In addition, despite the promise of this initial experience, patients’ and donors’ selection, route and frequency of administration, follow-up and tolerability all need better definition.

### 3.10. Direct Vigilance Modulation

Finally, direct modulation of vigilance, possibly in combination with ammonia-lowering drugs, is worthy of study.

There is low quality evidence suggesting a short-term beneficial effect of **flumazenil** on HE in patients with cirrhosis, with no influence on all-cause mortality [62]. It is reasonable that this drug may produce a transient improvement in severe overt HE (allowing the administration of additional treatment by mouth) and also revert any known or unrecognized previous benzodiazepine intake.

**Golexanolone** is a novel GABA-A receptor-modulating steroid antagonist under development for the treatment of cognitive and vigilance disorders caused by allosteric over-activation of GABA-A receptors by neurosteroids. It has been shown to restore spatial learning and motor coordination in animal models of HE [63] and to mitigate the effects of intravenous allopregnanolone in healthy adults [64]. A multi-center pilot RCT assessed safety, pharmacokinetics and efficacy of golexanolone in a small group of patients with cirrhosis [65]. Treatment seemed to improve neuropsychiatric performance, as demonstrated by a significant decrease in slow electroencephalographic frequencies.

The effects of **caffeine** on induced hyperammonaemia (amino acid challenge) was investigated in a study involving both healthy volunteers and patients with cirrhosis [66]. In healthy volunteers, the increase in ammonia levels due to the amino acid challenge was contained by both the administration of LOLA and that of caffeine. The administration of caffeine also resulted in a reduction in subjective sleepiness and in the amplitude of the EEG. Changes in ammonia levels, subjective sleepiness and the EEG were less obvious in patients. However, the timed administration of caffeine to top standard of care in patients with cirrhosis who also complain of sleepiness [67] may be both pleasant and useful.

### 3.11. Education

The provision of basic information on HE pathophysiology and simple hygienic and behavioural norms may have a significant impact on quality of life and HE recurrence in patients with cirrhosis. A 15 min educational session was administered to a small group of cirrhotic patients who had experienced at least one episode of overt HE [68]. This intervention was highly effective in increasing patients’ understanding of treatment of the condition and, ultimately, reduced the risk of HE recurrence over a period of one year.

### 3.12. Miscellanea

Finally, minocycline [69], ibuprofen [70], indomethacin [71], phosphodiesterase-5 inhibitors (i.e., sildenafil) [72], benzodiazepine inverse agonists [73], AST-120 [74], a microspherical carbon with a selective adsorbent profile for small molecules such as ammonia, and liposome-supported peritoneal dialysis [75] are all being tested in animal models and/or early phase clinical studies. For some, anecdotal, direct clinical experience is also available, especially where the tested drug has alternative indications.

### 3.13. Local Experience

The authors of this review run a daily, dedicated HE clinic within a tertiary referral hepatology centre [76]. While they adhere, as they have in this manuscript, to published treatment guidelines, it is their impression that routine clinical HE reality is complex, and its management more varied and in many ways more interesting and more satisfactory than one would expect. We utilise treatment to facilitate differential diagnosis, as patients with cirrhosis have multiple and often coexisting risk factors for neuropsychiatric impairment [77]: if HE is treated adequately one can rule out alternative diagnoses and/or establish their relative contribution to overall neuropsychiatric status. We use dietary changes, under strict and frequent monitoring, for highly recurrent and persistent HE, sometimes very successfully. We often resolve iatrogenic HE by simply easing strict dietary prescriptions (sometimes self-imposed) that have resulted in malnutrition and sarcopenia. We do not miss an opportunity to test experimental HE treatment strategies if co-morbidities allow us to do so safely. For example, we all recall a patient with shunt-related, persistent HE whose neuropsychiatric status drastically improved when he was started on a brief course of a non-steroidal anti-inflammatory drug because of backache. We are fortunate enough to have the facilities [78] to test the effects of treatment over time in a comprehensive fashion, and to tailor treatment accordingly. We regret that this clinical experience, partly because of inherent difficulties in describing/summarising it and partly because of publication policies in relation to case reports/case series, remains largely unpublished and transferred to colleagues by traditional and somewhat haphazard teaching and collaboration strategies. Finally, we have sometimes been able to test treatment in controlled conditions, i.e., by inducing hyperammonaemia and then attempting to lower it in different ways [66]. We find this to be an under-utilised but potent tool to better understand the pathophysiology of HE [79], and thus, improve its management.

**Table 1 jcm-10-04050-t001:** Available treatments (other than general principles, non-absorbable disaccharides/antibiotics) with some commentary on the evidence for them and/or tips for use.

Treatment Category	Treatment	Evidence or Tips for Use
Laxative	Polyethylene glycol	In the acute setting, when administration is safe by mouth or by naso-gastric tube
	L-Ornithine L-Aspartate	
Non ureic nitrogen scavengers	Sodium benzoate	Particularly useful when hyponatremia is also present
Non ureic nitrogen scavengers	Sodium phenylbutyrate	
Non ureic nitrogen scavengers	Glycerol phenylbutyrate	
Non ureic nitrogen scavengers	Ornithine phenylacetate	
Nutritional measures	Vegetarian/dairy diets	In patients with highly recurrent/persistent HE or those who are truly intolerant to animal protein Under tight monitoring to avoid lowering overall calorie/protein intake
Nutritional measures	Food intake distribution over the 24 h	3 snacks to top up the 3 main meals can be suggested to malnourished/sarcopenic patients If not tolerated, please insist on the late-evening snack, which is the most important
Nutritional measures	Branched-chain amino acids	Useful also as a late-evening snack and in association with vegetarian/dairy diets, to ensure adequate protein intake
Nutritional measures	Prebiotics, probiotics and symbiotics	Ecological approaches, such as increased soluble fibre intake (albeit not necessarily easy to obtain) most likely useful and free of side effects
Albumin		In patients with ascites Possibly also acting as a nutritional measure
Acetyl L-carnitine		
Fecal Microbiota Transplantation		
Direct vigilance modulation	Golexanolone	
Direct vigilance modulation	Caffeine	With attention to timing (max effect for 60–90 min after intake)
Miscellanea	Minocycline, ibuprofen, indomethacin, phosphodiesterase-5 inhibitors, benzodiazepine inverse agonists, AST-120, liposome-supported peritoneal dialysis	Experimental
Education		Limited evidence but reasonable approach, especially if slim and structured, for both patients and caregivers
Tertiary referral centre experience		Needs to be better and more formally described, published where possible and disseminated in a structured fashion

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
