# Peer review of "New Therapies of Liver Diseases: Hepatic Encephalopathy"

_jcm, 2021, doi:10.3390/jcm10184050_

Round 1

Reviewer 1 Report

The work by Chiara Mangini and Sara Montagnese is an up-to-date review presenting novel therapeutic substances and treatment strategies for hepatic encephalopathy (HE) in the course of liver cirrhosis.

HE is common one of the most serious complication of end-stage liver disease, affecting quality of life and combined with high hospitalization rate and shorten life expectancy. Therefore, the subject of the manuscript is actual, interesting, and clinically important from the clinical point of view.

Major comments:

A short introduction with definitions related to HE (i.e. types A-C, mild and overt HE) would make the text easier to understand for a reader non-specialized in hepatology.

The review aimed to present new drugs and therefore it only mentions well-grounded therapeutic molecules such as non-absorbable disaccharides and non-absorbable antibiotics. However short characterisation of these drugs would provide a more complex insight into the therapy strategies of HE. Moreover, in recent years several new studies, especially on ryfaximine were published (ex. Kang SH et al. Aliment Pharmacol Ther 2017; 46:845–855; Bannister CA, Orr JG, Reynolds AV et al. Clin Ther 2016; 38:1081.e4–1089.e4; Bajaj JS et al. Aliment Pharmacol Ther. 2015; Mullen KD et al. Clin Gastroenterol Hepatol. 2014 Aug;12(8):1390-7.e2; or Sidhu SS et al. Am J Gastroenterol 2011; 106:307).

Please add the information of the mechanism of action of polyethylene glycol (PEG).

Minor comments:

The manuscript is well written; however, the authors did not avoid some grammatical and stylistic errors. To ensure high quality of the text a language editing is highly recommended.

Shortcuts should be introduced after the first use of the whole word (ex. BCAA, LOLA).

Author Response

The work by Chiara Mangini and Sara Montagnese is an up-to-date review presenting novel therapeutic substances and treatment strategies for hepatic encephalopathy (HE) in the course of liver cirrhosis. HE is common one of the most serious complication of end-stage liver disease, affecting quality of life and combined with high hospitalization rate and shorten life expectancy. Therefore, the subject of the manuscript is actual, interesting, and clinically important from the clinical point of view.

Major comments:

A short introduction with definitions related to HE (i.e. types A-C, mild and overt HE) would make the text easier to understand for a reader non-specialized in hepatology.

We are grateful for the suggestion and have added the required paragraph to the Introduction of the amended paper.

The review aimed to present new drugs and therefore it only mentions well-grounded therapeutic molecules such as non-absorbable disaccharides and non-absorbable antibiotics. However short characterisation of these drugs would provide a more complex insight into the therapy strategies of HE. Moreover, in recent years several new studies, especially on ryfaximine were published (ex. Kang SH et al. Aliment Pharmacol Ther 2017; 46:845–855; Bannister CA, Orr JG, Reynolds AV et al. Clin Ther 2016; 38:1081.e4–1089.e4; Bajaj JS et al. Aliment Pharmacol Ther. 2015; Mullen KD et al. Clin Gastroenterol Hepatol. 2014 Aug;12(8):1390-7.e2; or Sidhu SS et al. Am J Gastroenterol 2011; 106:307).

We have added a small amount of information on the mechanisms of action of standard treatment and the suggested, additional references.

Please add the information of the mechanism of action of polyethylene glycol (PEG).

This has been added.

Minor comments:

The manuscript is well written; however, the authors did not avoid some grammatical and stylistic errors. To ensure high quality of the text a language editing is highly recommended.

The manuscript has been checked for grammar and style (these changes are not marked).

Shortcuts should be introduced after the first use of the whole word (ex. BCAA, LOLA).

These have been added.

Reviewer 2 Report

The present work aims to analyze the most adopted therapeutic strategies for the management of HC grade C.
In this regard, there are some elements that could be revised:
1. The methodology adopted for the inclusion of the works is not included: which period was taken into consideration? which databases have been adopted? which types of articles were included and which excluded? by how many people and how was the review of the articles carried out?
2. The quality and typology of the items considered is not clear. In this regard, it would be useful to evaluate the items under consideration with adequate checklists (CASP, Newcaslte-Ottawa, etc.)
3. A table for each therapeutic approach analyzed would make the concepts and evidence clearer
4. It is clear that we are dealing with a narrative review rather than a systematic review. However, it would be useful to specify it
5. The inclusion of management experience by the center can make the article more valuable.

The paper seems to be the introduction to a larger work (specialist book, degree thesis, etc.) and cannot be considered a complete work.
Greater attention to structure could lead to the development of a well-rounded narrative review.

Author Response

Reviewer 2

The present work aims to analyze the most adopted therapeutic strategies for the management of HC grade C.
In this regard, there are some elements that could be revised:

  1. The methodology adopted for the inclusion of the works is not included: which period was taken into consideration? which databases have been adopted? which types of articles were included and which excluded? by how many people and how was the review of the articles carried out?

As the reviewer correctly highlights, this is a narrative and not a systematic review. We included standard information on HE treatment (no formal literature search) and 2018-2021 information on novel therapies. These dates were chosen arbitrarily because author SM had previously completed a thorough review on HE treatment while working on the Italian guidelines on HE (Montagnese et al. Hepatic encephalopathy 2018: A clinical practice guideline by the Italian Association for the Study of the Liver (AISF). Dig Liver Dis 2019, 51:190-205). The 2018-2021 literature search was conducted on Pubmed using the terms hepatic encephalopathy plus: treatment, Polyethylene glycol (PEG), Branched-chain Amino Acids (BCAAs), L-Ornithine L-Aspartate (LOLA), nitrogen scavengers, Acetyl L-Carnitine , albumin, probiotics, Fecal Microbiota Transplantation (FMT), flumazenil, minocycline, ibuprofen, phosphodiesterase-5 inhibitors, indomethacin, benzodiazepine inverse agonists. A summary of the above information has been included in the Introduction of the amended manuscript. 

  1. The quality and typology of the items considered is not clear. In this regard, it would be useful to evaluate the items under consideration with adequate checklists (CASP, Newcaslte-Ottawa, etc.)

As indicated above, we were invited to provide a narrative review, thus formal checklists were not utilised.

  1. A table for each therapeutic approach analyzed would make the concepts and evidence clearer.

As indicated above, we were invited to provide a narrative review, and for most novel therapies there are only one or two available studies, thus a Table for each would seem somewhat redundant.

  1. It is clear that we are dealing with a narrative review rather than a systematic review. However, it would be useful to specify it.

We agree and we have added this piece of information in the Abstract and the Introduction of the amended manuscript.

  1. The inclusion of management experience by the center can make the article more valuable.

The paper seems to be the introduction to a larger work (specialist book, degree thesis, etc.) and cannot be considered a complete work.
Greater attention to structure could lead to the development of a well-rounded narrative review.

We are grateful for this suggestion and we have added a paragraph on local experience, together with four pertinent references. Structure has also been slightly modified but not to a significant extent – we hope these changes will be considered acceptable.

Round 2

Reviewer 2 Report

The work analyzes the delicate issue of the treatment of hepatic encephalopathy (HE).
All possible treatment levels are taken into consideration, carrying out a careful examination of the effects of each of the described treatments.
However, there are some points to consider:
1. The topic has a large number of scientific sources and is so widely known and treated: a narrative review adds little additional information to what is already known. A systematic review would be useful to take stock of the situation, analyzing concrete data related to the therapies. This task is made easier by the previous research work carried out by the authors for the drafting of the guidelines.
2. The quantity of references is such as to justify a systematic analysis of the data, rather than a narrative review: it would be necessary to reduce the refenreces to focus attention on certain scientific sources.
3. The absence of tables, figures or flowcharts makes it more difficult to understand the concepts present in the text.

Author Response

The work analyzes the delicate issue of the treatment of hepatic encephalopathy (HE). All possible treatment levels are taken into consideration, carrying out a careful examination of the effects of each of the described treatments.

We are grateful for this positive comment.

However, there are some points to consider:
1. The topic has a large number of scientific sources and is so widely known and treated: a narrative review adds little additional information to what is already known. A systematic review would be useful to take stock of the situation, analyzing concrete data related to the therapies. This task is made easier by the previous research work carried out by the authors for the drafting of the guidelines.

We completely agree. However, this is the type of manuscript we were invited to submit and at this stage we do not really have the resources and the time to invest in a systematic review which, we fully agree, would be a better and more useful endeavour than a narrative one.

  1. The quantity of references is such as to justify a systematic analysis of the data, rather than a narrative review: it would be necessary to reduce the refenreces to focus attention on certain scientific sources.

We have removed 7 redundant references (numbers 5, 6, 15, 16, 17, 23 and 24 of the previous version).

  1. The absence of tables, figures or flowcharts makes it more difficult to understand the concepts present in the text.

We agree and we have added a Table which will hopefully help cover also some of the issues raised in points 1 and 2 above.